# Genetic and Pathogenic Variability among Isolates of *Sporisorium reilianum* Causing Sorghum Head Smut

**DOI:** 10.3390/jof10010062

**Published:** 2024-01-12

**Authors:** Louis K. Prom, Ezekiel Jin Sung Ahn, Ramasamy Perumal, Thomas S. Isakeit, Gary N. Odvody, Clint W. Magill

**Affiliations:** 1USDA-ARS, Plains Area Agricultural Research Center, College Station, TX 77845, USA; 2USDA-ARS, Plant Science Research Unit, St. Paul, MN 55108, USA; ahn00079@umn.edu; 3Department of Agronomy, Agricultural Research Center, Kansas State University, Hays, KS 67601, USA; perumal@ksu.edu; 4Department of Plant Pathology and Microbiology, Texas A&M University, College Station, TX 77843, USA; thomas.isakeit@ag.tamu.edu (T.S.I.); c-magill@tamu.edu (C.W.M.); 5Department of Plant Pathology and Microbiology, Texas AgriLife Research Station, Corpus Christi, TX 78406, USA; gary.odvody@ag.tamu.edu

**Keywords:** sorghum, head smut, *Sporisorium reilianum*, genetic variation, pathogenicity, virulence pattern

## Abstract

*Sporisorium reilianum*, the causal agent of sorghum (*Sorghum bicolor* (L.) Moench) head smut, is present in most sorghum-producing regions. This seed replacement fungal disease can reduce yield by up to 80% in severely infected fields. Management of this disease can be challenging due to the appearance of different pathotypes within the pathogenic population. In this research, the genetic variability and pathogenicity of isolates collected from five Texas Counties was conducted. Due to the lack of available space, 21 out of 32 sequenced isolates were selected and evaluated for virulence patterns on the six sorghum differentials, Tx7078, BTx635, SC170-6-17 (TAM2571), SA281 (Early Hegari), Tx414, and BTx643. The results reveal the occurrence of a new pathotype, 1A, and four previously documented US pathotypes when the 21 isolates were evaluated for virulence patterns on the differentials. The most prevalent was pathotype 5, which was recovered from Brazos, Hidalgo, Nueces, and Willacy Counties, Texas. This pathotype was followed by 1A and 6 in frequency of recovery. Pathotype 4 was identified only from isolates collected from Hidalgo County, while pathotype 1 was from Burleson County, Texas. It appeared that the previous US head smut pathotypes (2 and 3) are no longer common, and the new pathotypes, 1A, 5, and 6, are now predominant. The phylogenetic tree constructed from the single-nucleotide polymorphism (SNP) data through the neighbor-joining method showed high genetic diversity among the tested isolates. Some of the diverse clades among the tested isolates were independent of their sampled locations. Notably, HS37, HS49, and HS65 formed a clade and were classified as 1A in the virulence study, while HS 61 and HS 66, which were collected from Nueces County, were grouped and identified as pathotype 5.

## 1. Introduction

Sorghum (*Sorghum bicolor* (L.) Moench) head smut, caused by *Sporisorium reilianum* (Kühn) Langdon and Fullerton (syns. *Sphacelotheca reiliana* (Kühn) G. P. Clinton, and *Sorosporium reilianum* (Kühn) McAlpine), is present in most sorghum-producing regions [1,2,3,4,5]. The fungus has two distinct populations: *S. reilianum* f. sp. *reilianum*, which infects sorghum, and *S. reilianum* f. sp. *zeae*, which causes head smut on maize [6,7,8,9]. In this study, pathogenicity, which refers to the ability to cause disease in the host, and virulence, which indicates either the degree or worsening of the disease [10], will be used interchangeably. The infection process begins when the teliospores of *S*. *reilianum* in the soil germinate in response to the germinating host seeds, and the pathogen develops dikaryotic hyphae that penetrate the meristematic tissue of the seedling [11,12,13]. However, the obvious phenotypic symptom arises later after colonization of the apical meristem so that at flowering, the pathogen replaces the seeds with masses of teliospores [4,13]. Globally, sorghum head smut is increasing due to several factors, such as the planting of susceptible cultivars, continuous mono-cropping, and variability within the pathogenic population [4,5,14,15,16,17]. In China, the mean incidence of head smut on sorghum was reported in the 1990s, ranging from 18 to over 80% across 58 thousand hectares [12]. A survey conducted in Western Kenya found that 73 to 75% of the farmers’ fields were infected with head smut [3]. Yield losses can reach 80% in severely infected sorghum fields [11,12,17]. In some sorghum-growing regions in China, losses ranging from 5 to 80% due to head smut were reported [5]. Significant yield losses were noted in sorghum fields in south Texas, USA, and in Tamaulipas and Ocotlan, Mexico, with high levels of head smut infestation [16]. Cultural practices such as the removal of infected plants and rotation may reduce the disease but are ineffective due to spore survival, as they may last up to 10 years in the soil, with occasional germination of spores at low rates when conditions are favorable [18,19]. Seed treatment may reduce head smut incidence; however, the most effective control host plant resistance strategy is the utilization of resistant sources [4,5]. Researchers have reported sources of resistance against several *S. reilianum* pathotypes. Sorghum lines, including B421, A232E, and BICS49, were shown to be resistant against pathotype 3 [17], while PI961515, PI961516, and PI961560 were resistant to pathotypes 1, 2, 3, and 4 [5]. Recently, 67 accessions from the sorghum association panel were evaluated against pathotypes 5 and 6. The results showed that PI656033 and PI656015 were resistant to pathotype 5 but exhibited a susceptible response when challenged with pathotype 6, while PI656090, PI564165, and PI642998 were susceptible to pathotype 5 h but resistant to pathotype 6 [16]. In the same study, Pi651492, PI656082, and PI533927 were resistant to both pathotypes. Pecina-Quintero et al. [20] studied the association between cytoplasm types (A_1_ and A_2_) among several hybrids and head smut resistance in field trials. The authors reported that hybrids with cytoplasm type A_1_ were more resistant to head smut.

Hence, the identification of molecular markers associated with host and race specificity and monitoring of the virulence pattern of the emerging new isolates are needed for the identification of genes controlling fungal pathogenicity. Although head smut is a disease of global economic importance, limited research has been conducted on the genomics and pathogenicity of *S. reilianum*. Genetic variability among 459 sorghum head smut isolates collected from the US, Mexico, and Niger was determined by Naidoo and Torres-Montalvo [6] using RFLP analysis, and the results showed low levels of genetic diversity within each population. Prom et al. [4] assessed the genetic diversity of 49 (44 from Texas, 2 from Uganda, 1 from Mali, and 3 maize isolates from Mexico) *S. reilianum* isolates using 16 amplified fragment-length polymorphism primer combinations and noted high genetic differences between isolates obtained from maize and those from sorghum. The work also grouped the Texas isolates collected from sorghum into four small clusters with ≥82% similarity using cluster analysis. Using 173 anonymous probes from two genomic libraries, a low level of variation was detected among 10 sorghum head smut isolates collected from China, Mali, Mexico, Uganda, and the USA [21]. Population structure analysis of 53 *S*. *reilianum* f. sp. *zeae* isolates from Mexico identified three genetic groups with low variation among the groups [22]. Until recently, four pathotypes (1, 2, 3, and 4) of the sorghum head smut pathogen were established by Frederiksen et al. [23] in the US based on their virulence patterns on four sorghum lines: Tx7078, SA281, Tx414, and TAM2571 (SC170-6-17). Dodman et al. [24] detected pathotypes 1 and 3 in Queensland, Australia, on field-grown cultivars/hybrids. Using one Chinese, one Indian, and three US lines, Zhang et al. [5] were able to establish four Chinese pathotypes: 1, 2, 3, and 4. When Chinese pathotype 4 was inoculated on the four sorghum differentials compiled by Frederiksen et al. [23], it infected SC170-6-17, while Tx7078 and SA281 were immune, and Tx414 was moderately resistant. Prom et al. [4] added a stable resistant line, BTx635, and susceptible BTx643 to the original set of four differentials by Frederiksen et al. [23] and documented two new US pathotypes, 5 and 6. Due to the variability in the head smut pathogen, there is a need to continue to monitor changes in the pathogenic population. Thus, the research was conducted with the objectives of characterizing the genetic diversity, determining pathogenic variability in 21 out of the 32 sequenced *S. reilianum* isolates collected in Texas, and relating the isolates with the genotypic single-nucleotide polymorphic (SNP) fingerprints.

## 2. Materials and Methods

*Head smut isolates*: Sorghum head smut isolates were collected from five Texas Counties: Brazos (HS122), Burleson (HS22 and HS81), Hidalgo (HS35, HS37, HS40, HS44, and HS47), Nueces (HS13, HS49, HS52, HS61, HS65, HS66, HS71, HS74, HS86, and HS107), and Willacy (HS109 and HS124). The location for HS130 was not recorded. All the isolates were collected in 2016 and 2017 growing seasons, except isolate HS35 in 2004 and HS13 and HS22 in 2011. 

*DNA extraction and phylogeny reconstruction of S. reilianum*: The MasterPure^TM^ Yeast DNA Purification kit (Biotechnologies) (Fisher Scientific, Pittsburg, PA, USA) was used to extract fungal DNA from 32 isolates of *S. reilianum*. These DNA extracts were subjected to restriction-site-associated sequencing (RAD-Seq) at the Genomics and Bioinformatics Service of Texas A&M AgriLife Research (TxGen, College Station, TX, USA). Using ILLUMINA technology (Illumina, San Diego, CA, USA), each isolate was sequenced from both ends of the restriction fragments after barcoding. The resulting reads, which ranged from approximately 780,000 to nearly 7 million per sample, were pre-screened, stripped of primer adaptor and barcode sequences, and aligned to sequenced contigs from a draft of the *S. reilianum* genome in GenBank (NCBI Accession: PRJEB19311 ID: 434910) using the CLC Genomics Workbench (v8). Tools in the same software were used to generate a collection of 16,382 bp SNP data from the 32 *S. reilianum* isolates. The CLC workbench software was employed due to the fact that it is user-friendly and an accurate tool for analyzing data for variant discovery [25]. Phylogeny reconstruction was carried out using Molecular Evolutionary Genetics Analysis Version 11 (MEGA11) software by the neighbor-joining method and the Maximum Composite Likelihood model with 10,000 bootstrap replications [26,27,28,29,30] based on the SNP data.

*Sorghum differentials*: the sorghum lines Tx7078, BTx635, SC170-6-17 (TAM2571), SA281 (Early Hegari), Tx414, and BTx643 are used regularly for identifying the resistance source, and virulence studies by Prom et al. [4] were used in this study.

*Greenhouse evaluation for race determination*: Eight seeds from each sorghum differential were planted in 3-gallon pots filled with potting soil (Metro Mix 200, Sun Gro Horticulture, Agawam, MA, USA) and, after germination, thinned to five plants per pot. The experiment was arranged in a completely randomized block design, with each differential replicated three times. The greenhouse experiment was allowed to run for reliable readings and to confirm the results in the secondary tillers, as there is always a chance for delayed disease infection following artificial syringe inoculation.

The inoculum preparation, inoculation method, and disease assessment were previously described by Prom et al. [16]. Individual smutted sorghum panicles were collected and placed in different paper bags and allowed to dry for 4 to 5 days. Teliospore were passed through a fine metal sieve to separate them from the debris and stored in sterile vials at 4 °C in the refrigerator until ready for use. Figure 1 shows an illustration of the inoculum preparation and the inoculation protocol. Briefly, sporidial colonies from germinating teliospores of the individual head smut isolates were placed in separate flasks containing potato dextrose broth. The flasks were placed on a rotary shaker set at 150 rpm for 4 d at room temperature to increase the number of the yeast-like spores. The sporidial suspension was filtered through four layers of sterilized cheesecloth into separate Erlenmeyer flasks and adjusted to a concentration of 1 × 10^5^ spores mL^−1^. The 18- to 20-day-old seedlings in each pot were inoculated below the apical meristem by injection of 0.5 to 1.0 mL of the sporidial suspension using a Precision Glide Needle # 22 G × 1 in. (Becton, Dickinson and Co., Franklin Lakes, NJ, USA) attached to a 30 mL hypodermic syringe. Plants were evaluated at the flowering stage for head smut infection. Plants with grains in the main tiller and no head smut symptoms were classified as resistant, while those with any of the characteristic head smut symptoms (Figure 2; Prom et al. [4]) were rated as susceptible. To confirm the reaction of those plants rated as resistant because the main tiller had healthy grain and no head symptoms, further evaluation was conducted by cutting the main tiller and allowing the side tillers in the ratoon crop to grow to the flowering stage. If the disease symptoms were expressed in the side tillers, the line was classified as susceptible. Side tillers on plants whose main tillers were cut and still without disease were classified as resistant. Thus, the reaction of the differentials was binary (resistant or susceptible).

## 3. Results

The genetic diversity of *S. reilianum* isolates was revealed through the phylogenetic analysis of 16,382 bp SNP data from 32 isolates collected in Texas. The phylogenetic tree constructed from the SNP data through the neighbor-joining method showed high genetic diversity among the tested population (Figure 3). This figure also shows all branches, and those higher than 50% were numbered. About half of the isolates did not group with any other isolate, indicating the high level of genetic diversity present within *S. reilianum*. As an example, isolates from Nueces County were scattered with high diversity.

*Pathotype determination*: All the sorghum head smut isolates used in this study were collected from Texas, US. Among the 32 isolates sequenced, 21 genetically diverse isolates were tested for their virulence patterns on the six differentials (Tx7078, BTx635, SC170-6-17 (TAM2571), SA281 (Early Hegari), Tx414, and BTx643). The results reveal the identification of a new pathotype designated as 1A (Table 1). Pathotype 1A was found in the Burleson, Hidalgo, and Nueces Counties. Other pathotypes identified in the study were 1, 4, 5, and 6. Pathotype 4 was identified in Hidalgo County, and pathotype 1 was detected only in Nueces County. The most frequently identified pathotype, 5, was detected in the Nueces, Hidalgo, and Willacy Counties but was most frequently recovered from isolates (HS13, HS61, HS66, HS74, and HS107) collected in Nueces County (Table 1). Pathotype 6 (HS40, HS52, HS122, and HS124) was found in the Brazos, Hidalgo, Nueces, and Willacy Counties.

## 4. Discussion

The world’s population is expected to reach around 9.1 billion inhabitants by 2050 and will require an increase in cereal production, including sorghum, to 3.0 billion tons for human consumption, animal feed, industrial needs, and other uses [31]. Due to the reliance of hundreds of millions of people in subtropical and semi-arid regions on sorghum for food, feed, and other uses because of its drought tolerance and adaptability in marginal lands, it is indispensable for global food security [16,32,33]. However, sorghum productivity and profitability can be impacted by several biotic stresses, including head smut caused by *S. reilianum* [16].

This study investigated the genetic characterization among the 32 *S. reilianum* isolates collected in various counties in Texas and the potential correlation with virulence patterns. The phylogenetic analysis revealed diverse clades among the tested isolates, independent of their sampled locations. HS37, HS49, and HS65 formed a clade and were classified as 1A in the virulence study, while HS 61 and HS 66, collected from Nueces County, were grouped and identified as pathotype 5. The genetic diversity in this study was evident as some of the isolates, such as HS13, HS74, HS81, and HS122, did not form a group with other isolates. *S. reilianum* requires the union of nuclei that differ in mating-type alleles for two different gene loci to infect and grow through the host in a dikaryotic phase followed by the formation of diploid teliospores at the time of heading [34]. These cycles of sexual reproduction may help preserve the high level of DNA-based diversity seen in this and similar pathogens. The study did not reveal if any markers were strictly associated with a particular pathotype. However, the isolates in clades could easily be explained by the fact that the sources were from relatively nearby fields and may also have somewhat reflected common cultivars (susceptible) grown in the region. Even though the infection process generally occurs through the soil at the seedling stage, there is typically sexual reproduction involved in every generation (even though many ‘crosses’ may involve products from one teliospore), meaning new combinations can readily occur.

Currently, there are different tools available for variant discovery and to generate phylogenetic trees. Drees et al. [35] constructed a high-resolution phylogenetic tree of *Pseudogymnoascus destructans*, a causal agent of white-nose syndrome, for bats using SNPs from whole-genome sequencing and microsatellites. The results showed that isolates collected from North America were less diverse than those from Europe, and it was concluded that the appearance of the pathogenic population in North America most likely originated from Europe instead of Asia. SNP-based analyses of population structure and a selective sweep of 25 isolates of *Sclerotinia sclerotiorum*, a cosmopolitan fungus collected from four continents, revealed two major populations and a low abundance of selective sweeps in the genome when compared to other fungal pathogen genomes [36]. Kulik et al. [37] analyzed 99 *Fusarium graminearum* isolates and 33 strains representing the known cryptic species of the pathogen using whole-genome sequencing (WGS). First, the author performed a phylogenomic analysis which revealed a species-specific clustering of all analyzed *F*. *graminearum* strains into a single clade by using the analyzed WGS information from individual strains to map the sequenced data against the reference genomes. According to the authors, this method is robust and provides a framework for typing *F*. *graminearum* through the web-based PhaME workflow available at EDGE bioinformatics [36]. At the time this study was conducted, there was no reference genome, so reads from each isolate were aligned and SNPs from each region were readily detected at any specific position.

Among the 21 isolates tested for virulence patterns in the greenhouse, 5 pathotypes were identified; however, it was not possible to connect a specific SNP to pathotypes. Further research is warranted to explore genes associated with the virulence pattern of *S. reilianum*.

Sorghum head smut pathosystem management can be challenging due to several factors, including variability within the pathogenic population [4,5,23]. This study explored the genetic and pathogenic variability in the head smut pathogen, S. reilianum. The phylogenetic analysis showed distinct genetic diversity among the 32 isolates collected from several counties in southern Texas. Due to space limitations, 21 isolates were selected for the virulence pattern study in the greenhouse. Over many decades, four *S. reilianum* pathotypes (1, 2, 3, and 4) using four sorghum differentials (Tx7078, SC170-6-17 (TAM2571), SA281 (Early Hegari), and Tx414) were recognized in the US [23]. Prom et al. [4], using the previous four differentials identified by Frederiksen et al. [23] and adding one stable resistant line, BTx635, and a susceptible line, BTx643, documented two new pathotypes, 5 and 6, in the US. The results from this current research identify four previously described US pathotypes (1, 4, 5, and 6) and one new pathotype designated as 1A (Table 1). The reason this new pathotype was described as 1A is that in Frederiksen et al. [23], US pathotype 1 was able to infect Tx7078, while the remaining three differentials, SA281, Tx414, and SC170-6-17, were resistant. The new US pathotype 1A was able to infect SC170-6-17, while Tx7078, SA281, and Tx414 were resistant. In this current study and the previous work by Prom et al. [4], US pathotypes 2 and 3 were not present in the head smut isolates collected in Texas, while US pathotypes 5 and 6 seemed to be increasing. Frederiksen [11] noted that a rapid shift in the head smut pathogenic population could occur if specific host varieties were planted. According to Frederiksen [11], US pathotype 4 from Burleson County evolved from US pathotype 1 and was unable to infect US pathotype 3 differential TX414. In contrast, US pathotype 4 from Wilson County developed from US pathotypes 1 and 3 was capable of infecting Tx7078, Tx414, and SC170-6-17 but not SA281 [10]. This indicates that the infestation or prevalence of pathotypes 2 and 3 in the pathogenic population in Texas could be limited due to either the removal of susceptible cultivars to these pathotypes or more resistant sorghum cultivars could be used. In this work, US pathotype 5 was the most prevalent, followed by 1A and 6. It seems that the previous US pathotypes, 1, 2, and 3, are becoming less virulent than the new pathotypes, 1A, 5, and 6. The current shift should be expected due to several other factors, including a shift in weather patterns and the appearance of these new and more virulent pathotypes due to climate change. In China, Zhang et al. [5] used five sorghum lines (Sanchisan from China; A_2_V4 from India; 961530, 961560, and TAM428 from the United States) and documented four Chinese pathotypes (1, 2, 3, and 4). Using the four sorghum differentials established by Frederiksen et al. [23], the four Chinese pathotypes were found to have different virulence patterns when compared with the four previous pathotypes (1, 2, 3, and 4) described in the US [5]. On the four sorghum differentials compiled by Frederiksen et al. [23], the Chinese pathotype 2, had the same virulence pattern as US pathotype 6, as documented by Prom et al. [4]. In the US, efforts are being made to identify sources resistant to pathotypes 5 and 6 as well as determine the inheritance of the resistance genes [16].

## Figures and Tables

**Figure 1 jof-10-00062-f001:**
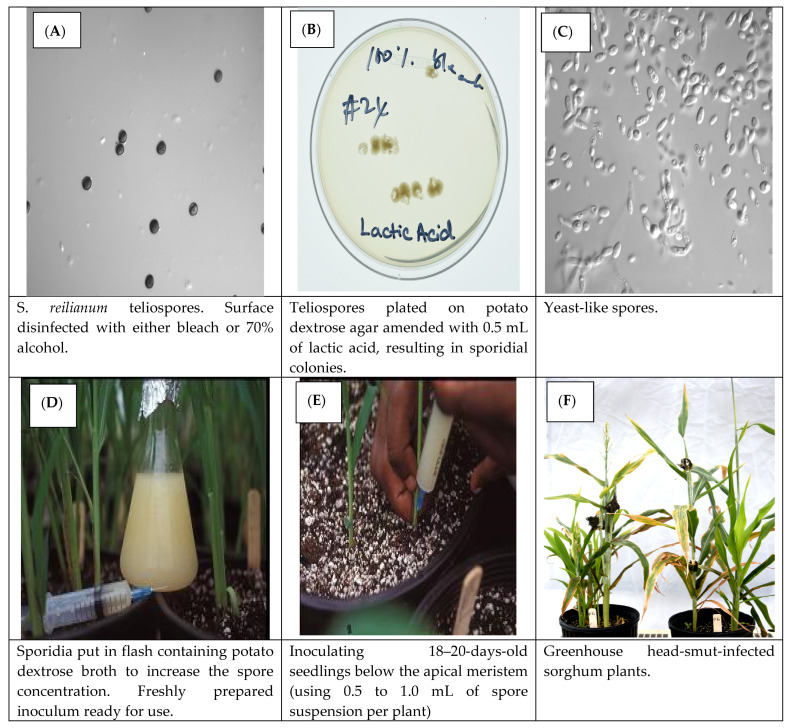
An illustration of the protocol for preparing inoculum and inoculating the sorghum seedlings.

**Figure 2 jof-10-00062-f002:**
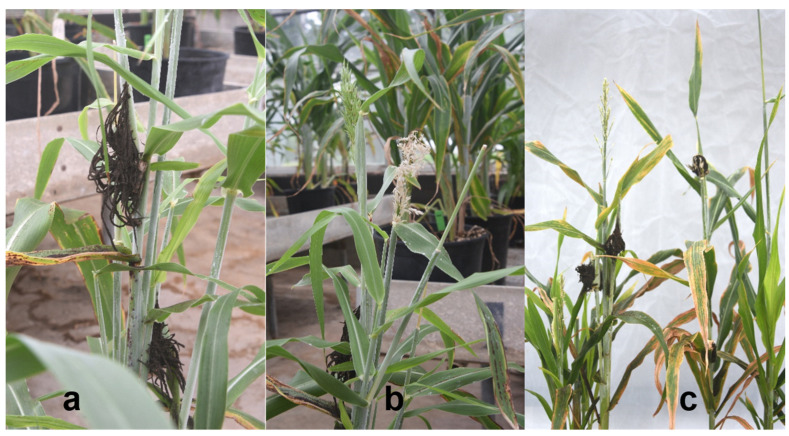
Head-smut-infected greenhouse-grown sorghum plants with different symptoms: (**a**) panicle completely replaced with sori, (**b**) blasted panicle, and (**c**) infection in secondary tillers.

**Figure 3 jof-10-00062-f003:**
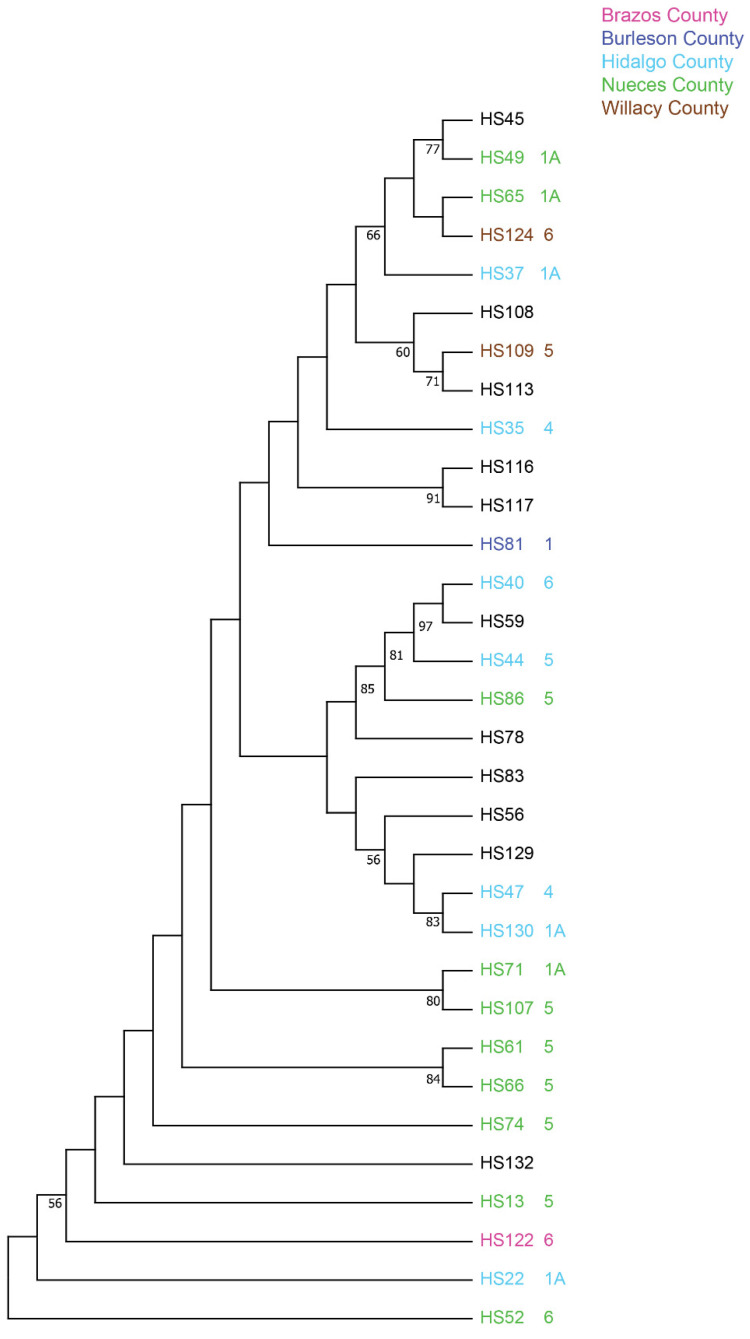
Phylogenetic tree constructed using SNPs data from 32 *S. reilianum* isolates. Branches with a bootstrap value greater than 0.5 are shown. The virulence pattern study included 21 isolates, denoted by different colors, with pathotypes indicated by numbers. The different colors represent the Counties in Texas where the head smut isolates were collected.

**Table 1 jof-10-00062-t001:** Sorghum head smut pathotype designations based on the virulence pattern of 21 isolates using six differentials.

Texas—County	Isolate	TX7078	SA281 (Early Hegari)	TX414	SC170-6-17 (TAM2571)	Btx635	BTx643	Pathotype
Nueces	HS13	S	R	R	S	R	S	5
Burleson	HS22	R	R	R	S	R	S	1A
Hidalgo	HS35	S	R	S	S	R	S	4
Hidalgo	HS37	R	R	R	S	R	S	1A
Hidalgo	HS40	R	R	S	S	R	S	6
Hidalgo	HS44	S	R	R	S	R	S	5
Hidalgo	HS47	S	R	S	S	R	S	4
Nueces	HS49	R	R	R	S	R	S	1A
Nueces	HS52	R	R	S	S	R	S	6
Nueces	HS61	S	R	R	S	R	S	5
Nueces	HS65	R	R	R	S	R	S	1A
Nueces	HS66	S	R	R	S	R	S	5
Nueces	HS71	R	R	R	S	R	S	1A
Nueces	HS74	S	R	R	S	R	S	5
Burleson	HS81	S	R	R	R	R	S	1
Nueces	HS86	S	R	R	S	R	S	5
Nueces	HS107	S	R	R	S	R	S	5
Willacy	HS109	S	R	R	S	R	S	5
Brazos	HS122	R	R	S	S	R	S	6
Willacy	HS124	R	R	S	S	R	S	6
Hidalgo	HS130	R	R	R	S	R	S	1A

## Data Availability

The data presented in this study are available upon reasonable request from the corresponding author.

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
