# Peer review of "Genetic and Pathogenic Variability among Isolates of Sporisorium reilianum Causing Sorghum Head Smut"

_jof, 2024, doi:10.3390/jof10010062_

Round 1
Reviewer 1 Report
Comments and Suggestions for Authors
Comments to the manuscript “Genetic and pathogenic variability among isolates of Sporisorium reilianum causing sorghum head smut” by Prom et al.
General comment
The submitted manuscript analyses the phylogenetic relationships of 32 isolates of the phytopathogenic fungal species Sporisorium reilianum obtained from five different Texas Counties (UAS). The phylogenetic tree was generated using SNPs from whole genomes obtained by using Illumina platform. Of these 32 isolates, 21 isolates were selected to determine the pathotypes by infection assays using six sorghum lines (differentials). Based on the obtained phylogeny the authors conclude that the genetic diversity of the studied isolates was high. The 21 isolates correspond to 5 different pathotypes, including the new 1A pathotype. Also, the results suggest that previously common USA pathotypes 2 and 3 have been replaced by the pathotypes 1A, 5, and 6.
The document is suitable for its publication in the Journal of Fungi after a mayor review. Below are some specific comments for the authors' consideration.
Specific comments:
1. Please provide a more detailed description of the pipeline or modules of the software used for SNPs detection between the genomes of the analyzed strains. For a robust phylogenetic analysis some works use core synapomorphic SNPs obtained from the core genome and after removing the duplicated/repeat regions (e. g. Drees et al. 2017. MBio. 8(6): e01941-17; Kulik et al. 2022. Frontiers in Microbiology. 13: 885978.). Also, adequate variant calling and quality filtering can be used to refine the SNPs used for phylogenetic analysis (e. g. Derbyshire et al. 2019. PloS One. 14(3): e0214201). Did you use some of such strategies to refine your SNPs data? Please clarify.
2. The phylogenetic tree obtained presents a main polytomy, which implies the lack of phylogenetic resolution to differentiate most of the isolates. This lack of resolution can be caused by a highly clonal population or by inappropriate phylogenetic reconstruction. If the analyzed isolates cannot be separated in robust dichotomic clades, such a result contradicts your assertion of high genetic diversity of the analyzed isolates. How do you interpret this lack of resolution? Please comment.
3. If your phylogenetic reconstruction is true (but see previous comment), how do you explain the clustering of isolates from different counties and pathotypes in the same clade (isolates HS22 and HS52; isolates HS13 and HS122)? Can such clusters reflect migration with subsequent pathotype differentiation? What other alternative hypothesis can be generated? Please comment on this regard.
Author Response
Reviewer 1
We would like to extend our gratitude to you your valuable comments and suggestions. The manuscript was amended accordingly.
REVIEWER 1.
- Please provide a more detailed description of the pipeline or modules of the software used for SNPs detection between the genomes of the analyzed strains. For a robust phylogenetic analysis some works use core synapomorphic SNPs obtained from the core genome and after removing the duplicated/repeat regions (e. g. Drees et al. 2017. MBio. 8(6): e01941-17; Kulik et al. 2022. Frontiers in Microbiology. 13: 885978.). Also, adequate variant calling and quality filtering can be used to refine the SNPs used for phylogenetic analysis (e. g. Derbyshire et al. 2019. PloS One. 14(3): e0214201). Did you use some of such strategies to refine your SNPs data? Please clarify.
Authors’ response: We used ‘well verified’ SNPs (this could be done visually as well as ‘electronically’ using the CLC package given in the methods. Since our primary goal was to see if any markers were strictly associated with a particular pathotype, which was not the case, we were not very concerned how the relationships turned out beyond seeing they were not in fact identical with would have suggest asexual (clonal) reproduction. The isolates in clades could easily be explained by the fact that the sources were from relatively nearby fields and may also have somewhat reflected common cultivars (susceptible) grown in the region. Even though the infection generally occurs through the soil, there is typically sexual reproduction involved every generation (even though many ‘crosses’ may involve products from one teliospore) meaning new combinations can readily occur.
- The phylogenetic tree obtained presents a main polytomy, which implies the lack of phylogenetic resolution to differentiate most of the isolates. This lack of resolution can be caused by a highly clonal population or by inappropriate phylogenetic reconstruction. If the analyzed isolates cannot be separated in robust dichotomic clades, such a result contradicts your assertion of high genetic diversity of the analyzed isolates. How do you interpret this lack of resolution? Please comment.
Authors’ response: The newly built tree differentiated the isolates you mentioned from one to the other, and we hope it resolved your concern. Also, variability within the pathogenic population exists in different sorghum growing regions.
- If your phylogenetic reconstruction is true (but see previous comment), how do you explain the clustering of isolates from different counties and pathotypes in the same clade (isolates HS22 and HS52; isolates HS13 and HS122)? Can such clusters reflect migration with subsequent pathotype differentiation? What other alternative hypothesis can be generated? Please comment on this regard.
Authors’ response: We still see the clustering of isolates from different Counties and pathotype in the same clade with the new tree (ex: HS65 and HS124). Since all the Counties are located in Southern Texas, it is not surprising that the migration with the isolates happened.
Reviewer 2 Report
Comments and Suggestions for Authors
Sporisorium reilianum is the causal agent of head smut on sorghum, causing large losses in various regions of the world. The current manuscript examines this important plant pathogen based on samples from Texas, USA. The research is interesting and provides significant results for science and sorghum disease management and therefore deserves publication in JoF. At this stage, however, the manuscript requires minor but important corrections and additions (see Remarks). Some fragments of the text are unclear because they contain certain omissions and inconsistencies. This applies especially to Materials and Methods.
Remarks
Line 19 it is '21 isolates were evaluated for virulence pattern’, however in Lines 91-95 only 20 isolates numbers are given. This needs to be explained.
Line 19 ‘against six sorghum differences’; in lines 113-115 are listed seven sorghum differentials that were used in this study. This is unclear and needs clarification. Probably BTx635 is mentioned twice
Line 32 in Keywords and in the text (e.g. lines 19, 28, 58, 59…) two definitions are used: pathogenicity and virulence. Since they are understood in different ways in plant pathology, it is advisable to devote a small paragraph to this aspect in the Introduction to explain how the authors define these concepts in their manuscript.
Line 91 'Head smut isolates' I see a certain problem with using the term 'isolates' (see Dictionary of the fungi, Kirk et al.). In phytopathology, the term isolate generally means fungal culture on agar medium, which, however, is very difficult for biotrophic fungi. You have collected isolates or fragments of sorghum with fungal spores? Here you should describe in detail what the samples looked like, under what conditions and where they were stored. This needs to be explained what is meant by isolate.
Lines 91-95 list 20 isolates, while line 102 lists 32 isolates. It is necessary to state where these additional isolates came from. These are major inconsistencies in the text.
Line 75 based….based - a small change is recommended
Line 125 The fungus was sampled mainly in 2016, the year the pathogenicity test was performed, now it is 2023. How long have the teliospores remained viable??
Line 126 it should be 'Prom et al. [15].'
Line 146 ‘from 32 isolates collected in Texas’ – please compare with the data in Lines 91-97 – this is unclear, requires clarification
Line 156-157 'six differentials (Tx7078, BTx635, SC170-6-17 (TAM2571), SA281 (Early Hegari), Tx414, BTx635, and BTx643) - that is six or seven differentials - this requires explanation. Probably BTx635 is mentioned twice
Line 199 please check if the text is correct
Line 231 A2V4 or A2V4 ??
Line 310 it should be ‘genetics of’
Line 310 Sporisorium reilianum should be in italic
Line 333 and line 336 Journals names should be in capital letters (for uniformity)
Line 320 it should be reilianum instead of Reilianum
Author Response
REVIEWER 2
We would like to extend our gratitude to you for your valuable comments and suggestions. The manuscript was amended accordingly.
Line 19 it is '21 isolates were evaluated for virulence pattern’, however in Lines 91-95 only 20 isolates numbers are given. This needs to be explained.
Authors’ response: Yes, it is 21 isolates. HS13 isolate was not noted. Now this isolate is highlighted in the Materials and Method section.
Line 19 ‘against six sorghum differences’; in lines 113-115 are listed seven sorghum differentials that were used in this study. This is unclear and needs clarification. Probably BTx635 is mentioned twice.
Authors’ response: Agreed. BTx635 was entered twice, and one entry is now deleted from the text.
Line 32 in Keywords and in the text (e.g. lines 19, 28, 58, 59…) two definitions are used: pathogenicity and virulence. Since they are understood in different ways in plant pathology, it is advisable to devote a small paragraph to this aspect in the Introduction to explain how the authors define these concepts in their manuscript.
Authors’ response: Agreed. Noted and highlighted in the INTRODUCTION section.
Line 91 'Head smut isolates' I see a certain problem with using the term 'isolates' (see Dictionary of the fungi, Kirk et al.). In phytopathology, the term isolate generally means fungal culture on agar medium, which, however, is very difficult for biotrophic fungi. You have collected isolates or fragments of sorghum with fungal spores? Here you should describe in detail what the samples looked like, under what conditions and where they were stored. This needs to be explained what is meant by isolate.
Authors’ response: Head smut pathogen can be cultured in media. Figure 1 gives an illustration of the protocol employed in this study.
Lines 91-95 list 20 isolates, while line 102 lists 32 isolates. It is necessary to state where these additional isolates came from. These are major inconsistencies in the text.
Authors’ response: Out of the 32 isolates sequenced, 21 were selected for the virulence pattern study in the greenhouse. Highlighted in the last sentence (INTRODUCTION section).
Line 75 based….based - a small change is recommended
Authors’ response: Sentence clarified by deleting “based.’
Line 125 The fungus was sampled mainly in 2016, the year the pathogenicity test was performed, now it is 2023. How long have the teliospores remained viable??
Authors’ response: Noted in the ‘INTRODUCTION’ section that the teliospores can survive up to 10 years.
Line 126 it should be 'Prom et al. [15].
Authors’ response: Agreed.
Line 146 ‘from 32 isolates collected in Texas’ – please compare with the data in Lines 91-97 – this is unclear, requires clarification.
Authors’ response: Agreed and clarification made in the text.
Line 156-157 'six differentials (Tx7078, BTx635, SC170-6-17 (TAM2571), SA281 (Early Hegari), Tx414, BTx635, and BTx643) - that is six or seven differentials - this requires explanation. Probably BTx635 is mentioned twice.
Authors’ response: Agreed that BTx635 was noted twice. One BTx635 was deleted.
Line 199 please check if the text is correct.
Authors’ response: Sentence was amended.
Line 231 A2V4 or A2V4 ??
Authors’ response: The 2 is a subscript as noted in the text (A2V4).
Line 310 it should be ‘genetics of’
Authors’ response: Noted.
Line 310 Sporisorium reilianum should be in italic
Authors’ response: Noted.
Line 333 and line 336 Journals names should be in capital letters (for uniformity).
Authors’ response: Noted.
Line 320 it should be reilianum instead of Reilianum
Authors’ response: Noted.
Reviewer 3 Report
Comments and Suggestions for Authors
The authors in their manuscript entitled “Genetic and pathogenic variability among isolates of Sporisorium reilianum causing sorghum head smut”, present the molecular characterization of 32 S. reilianum isolates and test the pathogenicity of 21 out of 32 selected isolates in different sorghum genetic backgrounds. The aim of research is clear, the methodological approach and implementation concerning the research presented are correct and the discussion of the results and subsequent conclusions support the main findings.
Revision of minor issues is suggested. In specific:
1. Line 99; delete the “2.1” prefix before the heading.
2. Line 194; Please revise or explain what is meant by “nearly half of the samples..”. I could count one third that do not couple (group).
3. Lines 198-199; It is difficult to understand the meaning of the sentence. Please revise.
4. Lines 205-206; There is no explanation why only 21 out of the 32 molecularly characterized isolates were selected for the pathogenicity tests. Please provide the argumentation and refer to it in the Summary and Introduction (Lines 84-87) parts as well.
Author Response
Reviewer 3.
We would like to extend our gratitude to you for your valuable comments and suggestions. The manuscript was amended accordingly.
REVIEWER 3
- Line 99; delete the “2.1” prefix before the heading.
Authors’ response: ‘2.1’ deleted.
- Line 194; Please revise or explain what is meant by “nearly half of the samples..”. I could count one third that do not couple (group).
Authors’ response: The sentence was clarified and highlighted in the DISCUSSION section.
- Lines 198-199; It is difficult to understand the meaning of the sentence. Please revise
Authors’ response: The sentence was amended and highlighted in the DISCUSSION section.
- Lines 205-206; There is no explanation why only 21 out of the 32 molecularly characterized isolates were selected for the pathogenicity tests. Please provide the argumentation and refer to it in the Summary and Introduction (Lines 84-87) parts as well.
Authors’ response: Reason for using 21 out of the 32 sequenced isolates is noted and highlighted in the DISCUSSION section.
Round 2
Reviewer 1 Report
Comments and Suggestions for Authors
Comments to the manuscript “Genetic and pathogenic variability among isolates of Sporisorium reilianum causing sorghum head smut” by Prom et al.
I appreciate the authors' response to my comments and suggestions.
However, the authors do not clarify my doubts about the way in which they obtained the SNPs used for phylogenetic analysis.
Respectfully, I still consider that they should clarify what "well verified" SNPs mean methodologically and not only cite the bioinformatic package used. Even more so because of the argument they mention to carry out a phylogenetic reconstruction, and the attained conclusion based on it.
If the SNPs were not generated from the core genome or repeated/duplicated regions were not eliminated (please see references from my first review), this may be the reason why they do not observe a marker associated with a particular pathotype.
On the other hand, the authors indicate that their phylogenetic tree clearly shows them the absence of clonality and sexual reproduction events in the small population of strains analyzed. However, more than 50% of the bifurcations in their new phylogenetic tree present bootstrap values below 0.5 (or 50%), so it does not have adequate statistical support. This is not commented on by the authors and does not allow robust conclusions about the clonality or not of the strains analyzed.
That is why they should be concerned with demonstrating that they carried out an adequate analysis and selection of SNPs for their phylogenetic reconstruction.
If after demonstrating that they made a bioinformatically adequate selection of the SNPs used, they obtain the same phylogeny of the figure 3, then they will be adequately confirming their argument and conclusions.
